# Bridging glacier and river catchment scales: an efficient representation of glacier dynamics in a hydrological model

Michel Wortmann<sup>1,2</sup>, Tobias Bolch<sup>3,4</sup>, Valentina Krysanova<sup>1</sup>, and Su Buda<sup>5</sup>

<sup>1</sup>Potsdam Institute for Climate Impact Research, Telegraphenberg A31, 14473 Potsdam, Germany

<sup>2</sup>Department of Geography, University College London, Gower Street, London WC1E 6BT, United Kingdom

<sup>3</sup>Department of Geography, University of Zurich, Winterthurer Strasse 190, 8057 Zuerich, Switzerland

<sup>4</sup>Institute for Cartography, Technische Universität Dresden, 01069 Dresden, Germany

<sup>5</sup>National Climate Centre, Chinese Meteorological Administration, No. 46, Zhongguancun South Street, Beijing, China

Correspondence to: Michel Wortmann (wortmann@pik-potsdam.de)

Abstract. Glacierised river catchments have been shown to be highly sensitive to climate change, while large populations depend on the water resources originating from them. Hydrological models are used to aid water resource management, yet their treatment of glacier processes is either rudimentary in large applications or linked to fully distributed glacier models that prevent larger model domains. Also, data scarcity in mountainous catchments has hampered the implementation of physically based

- approaches over entire river catchments. A fully integrated glacier dynamics module was developed for the eco-hydrological model SWIM (SWIM-G) that takes full account of the spatial heterogeneity of mountainous catchments but keeps in line with the semi-distributed disaggregation of the hydrological model. The glacierised part of the catchment is disaggregated into glaciological response units that are based on subbasin, elevation zone and aspect classes. They seamlessly integrate into the hydrological response units of the hydrological model SWIM. Robust and simple approaches to ice flow, avalanching,
- snow accumulation and metamorphism as well as glacier ablation under consideration of aspect, debris cover and sublimation are implemented in the model, balancing process complexity and data availability. The fully integrated is also capable of simulating a range of other hydrological processes that are common for larger mountainous catchments such as reservoirs, irrigation agriculture and runoff from a diverse soil and vegetation cover. SWIM-G is initialised and calibrated to initial glacier hypsometry, glacier mass balance and river discharge. While the model is intended to be used in medium to large river basins
- with data-scarce and glacierised headwaters, it is here validated in the data-abundant catchment of the Upper Rhone River, Switzerland and the data-scarce catchment of the Upper Aksu River, Kyrgyzstan/NW China.

# 1 Introduction

plagued with complications and uncertainties (Klemeš, 1990; Schaefli, 2005; Pellicciotti et al., 2012). Strongly heterogeneous

processes such as glacier dynamics, orographic precipitation and permafrost are paired with low observation densities, often resulting in severe data scarcity for hydrological modelling. Glaciers have been of particular concern, as their evolution under a changing climate may have significant consequences for downstream water resources (Huss et al., 2008; Immerzeel et al.,

Hydrological modelling and hydrological climate change impact assessments of mountainous and glacierised catchments are

2010; Bolch et al., 2012). Representing long-term glacier dynamics in a general-purpose hydrological catchment model has so far been limited (Naz et al., 2014).

Water management of glacierised catchments relies on hydrological models that estimate glacier melt contribution to river discharge of a given glacier cover, often referred to as glacio-hydrological models. Data gaps are overcome by spatial inter-

- polation and integration as well as empirical parametrisations. There is a range of conceptual, semi-distributed models with a long history that incorporate glacier melt successfully and are mainly based on the Degree-Day approach (Quick and Pipes, 1977; Schaefli, 2005; Hagg et al., 2007; Duethmann et al., 2013; Hock, 2005). They perform well over 1–10 year time scales, while being parametrised and calibrated to specific mountainous catchments. There are also some fully distributed and often more physically based models with resolutions of 25–300 m (Finger et al., 2011; Immerzeel et al., 2014; Huss et al., 2010b;
- Dickerson-Lange and Mitchell, 2014). They are spatially more explicit with more parameter redundancy and higher computational demands. Processes are implemented with more physical meaning, such as the full energy balance at the glacier surface for mass balance accounting. This class of models, however, suffers most from data scarcity, often leading to worse validation results than the former, due to the reliance on driving data (e.g. radiation, albedo etc.) that is not easily interpolated to the model domain.
- In addition to glacier mass balance modelling, only a few hydrological models consider ice dynamics, i.e. the lateral redistribution of ice down-slope under the force of gravity. For example, the fully distributed GERM model updates glacier cover annually using an empirical parametrisation of ice thickness changes (Huss et al., 2010b). Some two-dimensional glacier models (without catchment hydrology) have been developed to simulate glacier mass balances and ice dynamics from the glacier to the regional scale (Vieli, 2015; Clarke et al., 2015; Rowan et al., 2015). Fully integrated glacio-hydrological catchment models,
- however, are rare with the pioneering exception by Naz et al. (2014). They use the shallow ice approximation to evolve glacier surfaces in response to a full energy balance mass balance model and a comprehensive hydrological model at a resolution of 300 m, albeit a catchment of only 422 km<sup>2</sup> in size. Few semi-distributed, conceptual models consider ice dynamics as part of the glacier modelling, but glaciers are only represented as fractional coverage of elevation bands, neglecting the complex terrain in their mass balance and ice flow calculations (e.g. Uhlmann et al., 2013).

Short-comings of current glacio-hydrological models for long-term climate change impact assessments can be loosely divided into problems of a) integration and b) scale:

*Integration.* Most existing glacio-hydrological models have no or only a simple representation of the remaining catchment hydrology, as it remains a less important factor in small glacierised catchments. However, there is often a considerable distance between the glacierised part of a catchment and the locations where water becomes a socioeconomic and ecological resource.

It is in these locations where long-term hydrological observations are recorded that are needed for model calibration. The distance from the glaciers increases the catchment size, increasing the need for more accurate descriptions of the diverse landscape hydrology (e.g. vegetation, groundwater, irrigation agriculture and reservoirs). This is particularly important in long-term studies with drastic glacier changes: Glacier shrinkage exposes more area to solely hydrological processes, while precipitation increase promotes glacier growth but also more runoff in lower-laying areas of the catchment. Similarly, loosely

coupled approaches often simulate glacier mass balances and snow cover separately, leading to inconsistencies in the modelling chain.

*Scale*. Only fully integrated and mostly physically based models have been used to model both glacier evolution and catchment hydrology. Besides their high demand for driving data, they are constrained to relatively small catchments (a few 10s to

- a few 1000s of square kilometre in size). This is mainly due to their fully distributed nature and the related grid discretisation. In most cases, these models include a computationally intensive, two-dimensional finite-difference approach to ice flow, model runtimes increase drastically with finer resolutions, larger catchments and more sophisticated numerical solutions. The grid resolution is dictated by the complex terrain or the smallest glacier area that is intended to be represented, which puts the maximum grid size to several 100s of meters (Immerzeel et al. (2014) use 500m as the largest found in the literature). While
- higher resolutions are necessary for the representation of glacier processes, they are unnecessary for hydrological processes in large catchments with sparse observation data.

For long-term assessments of large (partially) glacierised catchments and full mountain ranges, an efficient model is required that integrates a complete description of the catchment hydrology with glacier mass balance and ice dynamics modelling (Naz et al., 2014).

This work's aim is to integrate glacier dynamics and mass balance processes into a general and process-based hydrological model to aid long-term and integrated climate change impact assessments. This will fill the gap between distributed small-scale glacio-hydrological models and large-scale hydrological assessments that ignore or strongly simplify ice dynamics.

In this pursuit, the semi-distributed, eco-hydrological Soil and Water Integrated Model (SWIM Krysanova et al., 1998) was extended by a glacier dynamics module (subsequently called SWIM-G), that includes the important glacier processes. It is

20 tested and validated in the data-scarce Upper Aksu catchment as well as the 'data-abundant' Upper Rhone catchment. While its intended use is primarily for data-scarce catchment, the Upper Rhone catchment serves as a validation case study and to contrast advantages and disadvantages both data conditions offer.

# 2 Methods

The model integration presented here is based on the idea that proven concepts in hydrological modelling exist (Peel and Blöschl, 2011), while glaciological models are highly specialised and not readily transferable to the catchment scale. Recent advances in glaciological modelling and the emergence of accurate, catchment and region-wide glacier outlines and mass balances (e.g. Fischer et al. (2014, 2015) for Switzerland, Pieczonka and Bolch (2015) for the Central Tien Shan) enable catchment-wide modelling on the glacier scale. In the following, the hydrological model SWIM is briefly outlined followed by more detailed descriptions of the newly implemented glacier processes. The calibration and validation strategy including the 20 two case study catchments are described at the end of this section.

two case study catchments are described at the end of this section.

# 2.1 The Soil and Water Integrated Model (SWIM)

The Soil and Water Integrated Model (SWIM) is a semi-distributed, process-based, ecohydrological model (Krysanova et al., 1998) with its origins in the semi-distributed model SWAT (Arnold et al., 1993). It was initially developed for long-term climate change impact assessments for medium to large river basins, but has since been developed into a fully integrated ecohydro-logical model encompassing a number of hydrological and water management processes for both water availability and water quality assessments Hattermann et al. (2011); Huang et al. (2010); Liersch et al. (2012); Koch et al. (2013). Krysanova et al. (2015) provides an overview of the hydrological processes considered and recent advances in its development.

An extended degree-day method is used to simulate snow melt (Huang et al., 2013a, b). It includes a continuous description of ice and water content in the snowpack as well as refreezing and metamorphism according to the approach of Gelfan et al.

- (2004). As it relies on accurate mean daily temperature, 100 m elevation bands are used to split hydrotopes and to adjust the subbasin mean temperature to the hydrotope elevation by a lapse rate that is catchment specific. The lapse rate varies between -0.3 °C/100 m for humid condition and up to -0.9 °C/100 m for dry conditions. Precipitation falls as snow if mean temperature falls below a threshold  $T_s$  and melts if it exceeds a threshold  $T_m$  via the degree-day method (Hock and Holmgren, 2005). Both thresholds are subject to calibration but are generally well-confined to  $\pm 3 \text{ °C}$ . The snow module of SWIM provides the main
- input to the newly developed glacier module, which is described in the following sections.

# 2.2 Spatial disaggregation of glaciers

SWIM is a semi-distributed, hydrological model with three levels of disaggregation: the basin, subbasins and the hydrotopes. The hydrotopes subdivide the subbasins typically by unique combinations of land cover, soil class and elevation band, but this can be refined by other variables. They provide an adaptive spatial unit depending on the process scale and available data.

- Taking on this proven hydrological concept, they are here used to represent glaciers. The hydrotopes are used to represent the complex mountainous terrain that determines glacier geometry and distribution by considering slope and aspect classes as well as elevation zones. This type of terrain abstraction is common in geomorphology with established threshold values and classification methods (Bishop et al., 2003; Cronin, 2000; Rasemann et al., 2004), while it is here used with the focus on glacier properties.
- In the potential glacier region of the river basin, the hydrotopes are unique combinations of three terrain classifications that are derived from the DEM: *a*) a valley and hillslope class (using a slope threshold), *b*) elevation zones with small intervals in valleys and larger intervals on hillslopes, and *c*) four, regularly spaced aspect classes on hillslopes only. The unique combinations produce a noisy map that needs to be cleaned with a minimum area threshold and successive neighbourhood filling (Figure 1).
- Typical elevation zones in hydrological models vary between 20–200 m (Lindström et al., 1997). The variable elevation zone intervals in valleys and hillslopes stems from the desire to have equally sized spatial units. Typical slope values for the two classes should thus govern the choice of intervals. A factor of 10 between valley and hillslope intervals, for example, will lead to equal downslope distances with typical slope angles of 3.9° and 34°. Distinguishing between slope aspect is important to

subdivide elevation zones. The aspect classes break these into distinct hillslope units that are more representative of glacial hillslopes than an entire elevation zone and distinguish glaciers with different exposure.

The slope threshold, the elevation intervals, the number of aspect classes and minimum cleaning area are threshold values that can be adapted to the desired level of terrain descretisation and are also dependent on the resolution of the DEM. Here, a slope threshold of  $< 15^{\circ}$ , elevation zones of 40 m in valleys and 400 m on hillslopes, four aspect classes and a minimum area of

slope threshold of  $< 15^{\circ}$ , elevation zones of 40 m in valleys and 400 m on hillslopes, four aspect classes and a minimum area of  $0.5 \text{ km}^2$  is used here. The minimum area threshold limits the model to glaciers larger than this threshold. These representative units resolve the glacial systems of the catchment as well as the hillslopes contributing to glacier accumulation.

Other spatial attributes relevant to the hydrological model are mapped onto the spatial structure of the glacier units, i.e. for each hydrotope the dominant land cover and soil class are used. As soil inventories in mountainous areas mainly apply to valleys (alluvial fans, plateaus) and the hillslopes are mainly composed of bare rock and extremely shallow soils, the soil depth

10 valleys (alluvial fans, plateaus) and the hillslopes are mainly composed of bare rock and extremely shallow soils, the soil depth on the hillslope units is reduced to 300 mm in line with typical soil depths in steep terrain (Dietrich et al., 1995; Heimsath et al., 1999).

## 2.3 Glacier formation and accumulation

All snow processes are governed by the existing snow module, this includes the description of ice and water content of the snow pack and melt is calculated accordingly. If at the end of the ablation season (defined as the last day of September) snow is left in the hydrotope, it turns into ice if it exceeds the critical height  $H_c$ , above which ice flow occurs.  $H_c$  is dependent on both slope  $\alpha$  and shear stress  $\tau_s$ , the force the ice needs to overcome to deform under its own weight. Although shear stress varies widely between glaciers and regions, a global average of  $10^5$  Pa is widely accepted (Cuffey and Paterson, 2010).  $H_c$  [m] is determined by the equation:

$$20 \quad H_c = \frac{\tau_s}{\rho \cdot g \cdot \tan(\alpha)} \tag{1}$$

with glacier ice density  $\rho$  (900 kg m<sup>-3</sup>), gravity g (9.8066 m s<sup>-2</sup>) and slope angle  $\alpha$  [°].

# 2.4 Ice flow

The routing between the glacier units is calculated similarly to the subbasin routing, i.e. according to the flow direction given in the DEM. Ice flow occurs if the critical height  $H_c$  is exceeded; if the thickness decreases below  $H_c$ , the ice area of the unit

is proportionally decreased to simulate slow terminus recessions. Figure 2 shows the routing between the glacier units in a single subbasin and a valley cross-section of three units. The flow volume  $Q_i$  [m<sup>3</sup> w.e. a<sup>-1</sup>] is based on Glen's Flow Law and the adaptation suggested by Marshall et al. (2011):

$$Q_i = \chi \cdot A_u \cdot H^5 \cdot \tan(\alpha)^3 \tag{2}$$

with area of the glacier unit  $A_u$  [m<sup>2</sup>], glacier thickness H [m w.e.], slope  $\alpha$  [°] and the rheology term  $\chi$  [m<sup>-4</sup>a<sup>-1</sup>] that is subject 30 to calibration. The flux  $Q_i$  is routed to the next glacier unit, but is constrained to the volume above the critical height.

5

20

To account for more accurate glacier area changes that in turn have a strong impact on catchment wide glacier melt, the glacier critical height is maintained if melting occurs while the glacier is at this level and instead the fraction of glacier area is reduced, as illustrated in Figure 2b. This simulates the gradual recession of a glacier front up-slope, exposing a decreasing area to melting after the glacier falls below the critical height. The vertical frontal area of the glacier snout needs to be accounted for to avoid the glacier unit to shrink indefinitely without ever disappearing completely. This area is approximated by the glacier height and the squareroot of the unit area. The glacier area of the glacier unit subject to melting  $A_m$  [m<sup>2</sup>] is given by:

$$A_m = A_u \cdot \frac{H}{H_c} + H_c \cdot \sqrt{A_u} \tag{3}$$

#### 2.5 Avalanching

Avalanching represents a more rapid form of snow and ice redistribution as the majority of the snow or firn column is removed
and transported to down-slope. The avalanche-prone areas are identified by a simple slope threshold that is physically based and well constrained to a range of 35–45° (Schweizer et al., 2003) and should be adapted to the observed glacier hypsometry and distribution. If the snow and glacier height exceed the critical height, the snow is accumulated on the remaining fraction of the glacier unit or if the avalanche proportion is greater than 90%, all snow is transported down-slope to the next glacier unit. This upper threshold is needed for numerical stability to avoid large snow masses 'piling up' on small fractions of the glacier

# 2.6 Glacier melt

The well-tested Degree-Day approach is implemented to simulate glacier melt, as temperature is the least uncertain available climate variable (Hock, 2003). Glacier melt is then collected in a linear reservoir together with liquid precipitation over the glacier, is subject to evaporation and released as glacier discharge  $Q_g$  with a delay described by the residence time. This is to simulate the water storage capacity of glaciers and the observed delay of glacier discharge after intensive melting periods (Cuffey and Paterson, 2010). The following equations describe the glacier melt  $M_g$  and water outflow  $Q_w$  from the linear reservoir  $V_w$ :

$$M_g = \begin{cases} \delta_g \cdot T & \text{if } T > 0 \text{ and } H_s = 0\\ 0 & \text{otherwise} \end{cases}$$
(4)

with the Degree-Day factor  $\delta_g \,[\text{mm}\,^\circ\text{C}^{-1}\,\text{d}^{-1}]$ , daily mean temperature  $T \,[^\circ\text{C}]$ , glacier  $H_g$  and snow height  $H_s$ .

$$25 \quad \frac{\delta V_w}{\delta t} = M_g + P_l - E - Q_w \tag{5}$$

$$Q_w = \frac{V_w}{t_r} \tag{6}$$

where evaporation E [mm]; liquid precipitation  $P_l$  and residence time  $t_r$  [d] ranges between 1–10 days and may be calibrated, for example, using individual melt events without rain. Potential evaporation is calculated by the Priestly-Taylor approach, the

standard in SWIM. The glacier water outflow is then subject to the same infiltration and surface runoff processes as liquid precipitation. The valley sediment is described by a highly permeable soil, which saturates quickly resulting in high rates of surface runoff.

Two processes are considered that alter the melt rate over space and time: a) slope aspect and terrain shading (Section 2.8) and b) supraglacial debris cover (Section 2.9). Both processes have been shown to have a significant influence on glacier melt and in turn the spatial distribution of glaciers over longer time periods. Although their governing processes are highly complex, two simple approaches are used to approximate their effect on Degree-Day melting rates, which are spatially distributed.

# 2.7 Sublimation

In most glacieriesed regions sublimation from the glacier is considered a negligible factor, with rates often far below the error of accumulation rates (Hock and Holmgren, 2005; Gascoin et al., 2011). This is mainly due to the fact that sublimation consumes 8.5 times as much energy (latent heat of sublimation:  $2.838 \times 10^6 \, \text{J kg}^{-1}$  versus latent heat of fusion:  $0.334 \times 10^6 \, \text{J kg}^{-1}$ ). In dry and high elevation zones, however, the proportion of energy consumed by sublimation rises to significant levels, suppressing melt rates and meltwater runoff as a result (Zhang et al., 2006; Mölg et al., 2009). High wind speeds and large vapour pressure deficits (or low relative humidity) favour sublimation and are common in high elevations. Modelling day-to-day variations in

sublimation rates is only possible with a full energy balance model. However, knowing approximate average ratios of energy used for sublimation, allows the coupling of sublimation with melting. Assuming ablation A is made up of sublimation S and melting M, the energy balance with a sublimation ratio  $\beta$  can be described as follows:

$$M = E \cdot \frac{1 - \beta}{L_f} \tag{7}$$

$$20 \quad S = E \cdot \frac{\beta}{L_s} \tag{8}$$

with the total available energy  $E [J kg^{-1}]$  and the latent heat of fusion  $L_f [J kg^{-1}]$  and of sublimation  $L_s [J kg^{-1}]$ . Using the Degree-Day approach from Equation (4), M can be replaced to solve for E as follows:

$$E = \delta_g \cdot T_+ \cdot \frac{L_f}{1 - \beta} \tag{9}$$

Using Equation (8), sublimation can be described by:

$$25 \quad S = \delta_g \cdot T_+ \cdot \frac{\beta \cdot L_f}{(1-\beta) \cdot L_s} \tag{10}$$

This allows to include sublimation from glaciers using the proven Degree-Day factor approach while only adding a single parameter, that can be estimated from general climatic conditions and sparse energy balance studies. Low observed or calibrated Degree-Day factors are also an indication for high proportions of energy used for sublimation (Zhao et al., 2006; Winkler et al., 2009).

# 2.8 Slope aspect and terrain shading

Slope aspect and terrain shading reduce the amount of short-wave solar radiation a glacier area receives, which is the predominant driver of glacier melt (Paul, 2010). A first order approximation of this variability is given by the potential sunshine duration per day a slope receives ignoring clouds, a variable readily inferable from a DEM. Although clear-sky solar radiation would provide a more accurate variable (as used in other models, e.g. Huss et al., 2008), it requires additional calibration parameters. Hours of sunlight are computed for both the summer  $h_s$  and winter  $h_w$  solstice and interpolated for all days in between with a sinus curve (the HBV-ETH model uses a similar sinusoidal differentiation of the Degree-Day factor, but with empirical boundaries, Hock and Holmgren, 2005). The basin-wide Degree-Day factor  $\delta_q$  is localised by linear scaling as  $\delta_i$ :

$$h_i = h_w + \frac{h_s - h_w}{2} \cdot \left(1 + \cos\frac{2\pi \cdot i}{365}\right) \tag{11}$$

5

$$\delta_i = \delta_g \cdot \frac{h_i}{12} \tag{12}$$

where *i* are the days since the winter solstice, sun hours on day *i*, on the summer and winter solstice are  $h_i$ ,  $h_s$ ,  $h_w$ . Although potential sun hours neglect cloud shadowing and the fact that melting is also driven by turbulent heat flux and defuse radiation, it provides an efficient method to vary the melt rate over complex terrain without introducing additional parameters, while the Degree-Day approach allows to calibrate the other melt terms implicitly.

#### 2.9 Debris cover

A supraglacial debris cover has long been shown to first increase glacier ablation up to a thickness of a few centimetres and then significantly decrease ablation (Bozhinskiy et al., 1986; Nicholson and Benn, 2006). The initial increase in melting is caused by the decreased albedo of debris and subsequent thermal conductivity to the glacier ice. This effect, however, is rapidly decreased

20 by the thermal shielding effect of debris layers thicker than a few centimetres. Observing the initial increase has been difficult and including the effect in modelling would require estimating debris thickness with errors smaller then the threshold thickness. Since this is beyond the precision of the model, only the decreasing effect of such a debris layer is considered here.

Several in situ studies have linked debris cover to subdebris ablation rates using a negative exponential relationship (Mattson, 1993; Nicholson and Benn, 2006, e.g. in the Himalaya and several other regions). Considering the measured daily mean

temperature, the ablation can be expressed in terms of Degree-Day factors, which vary between region and glacier. One such study has been conducted on the largest glacier of the Tien Shan, the heavily debris-covered South Inylchek glacier, Kyrgyzstan (Hagg et al., 2008). They find the following equation to describe the subdebris melt factor  $\delta_d$  with increasing debris cover thickness with a correlation of 0.94:

$$\delta_d = \delta_i \cdot e^{-0.0572 \cdot H_d} \tag{13}$$

with the clean ice melt factor  $\delta_i \, [\text{mm} \,^\circ \text{C}^{-1} \, \text{d}^{-1}]$ , and the debris thickness  $H_d \, [\text{cm}]$ . This empirical relationship is employed in the case of the Upper Aksu model, which includes the South Inylchek glacier in its catchment. The clean ice Degree-Day factor is subject to calibration.

As delineating the spatial distribution and estimating the thickness of debris cover over an entire river catchment is near to impossible, let alone knowing its development in the future, a dynamical approximation of the glacier cover was implemented.

- 5 impossible, let alone knowing its development in the future, a dynamical approximation of the glacier cover was implemented. Supraglacial debris has several origins; deposition of colluvial material, emergence of subglacial moraines and by melt out of englacial debris are the main processes involved (Bolch, 2011). While the first two processes are highly local processes and glacier specific, englacial debris melt out is the only one mainly driven by meteorology and universally applicable to a wider region (with varying intensity between regions). To simulate the evolution of debris produced by this process, an englacial
- 10 debris concentration approach is implemented (previously proven by Bozhinskiy et al. (1986) in a more complex form). While snow accumulation decreases the concentration, melting increases it and ice flow 'dilutes' the downstream concentration with the one upstream.

An assumed initial debris concentration  $C_{init}$  is altered by melting and accumulation in a glacier unit with the specific debris concentration  $C_g$  according to the following equation:

$$15 \quad \frac{\delta C_g}{\delta t} = C_g \cdot \left(1 + \frac{M_g - H_s}{H_g}\right) + \left(C_u - C_g\right) \cdot \frac{H_q}{H_g}, \quad C_{init} \le C_g 

Where meteorological information has to be extrapolated into great distances both horizontally and/or vertically, a method to correct for orographic precipitation is paramount to the accurate modelling of both the glaciers and the catchment hydrology (Immerzeel et al., 2014; Stisen et al., 2012). Most studies use linear gradients to vary precipitation with elevation over a complex terrain with typical ranges of  $0.05-0.5 \% \text{ m}^{-1}$ . However, representing variance of precipitation as a linear function of elevation is inherently local, highly variable over varying altitude ranges and generally unsuitable for elevations below the reference altitude. For the data-scarce Upper Aksu catchment in this study, precipitation is corrected by a function of altitude taking account of varying gradients and an eventual decrease at very high elevations. The correction factor  $f_c$  in Equation (16) remains 1 over lower laying elevations for which observations are available, but increases exponentially up to a maximum gradient *a*. It then reduces the gradient until a maximum correction *c* at altitude *m* is reached and decreases again at higher

10 elevations thereafter.

$$f_c(z) = (c-1) \cdot \exp\left[-\left(\frac{a}{c-1}\right)^2 \cdot (z-m)^2\right] + 1$$
(16)

This is a more dynamic approach than the combination of two linear functions proposed by Immerzeel et al. (2012). c is effectively the greatest correction applied, while the altitude m is the physical limit of the atmosphere to lose more moisture. For the Tien Shan, Aizen et al. (1995) provide approximate values for all three parameters of the correction function.

15

5

Since the Upper Rhone catchment has a much higher density of meteorological stations, the above form of altitude-dependent correction was not necessary. Instead, all available precipitation data were interpolated via the Inverse-Distance-Weighting method and corrected by factors published in the Swiss Hydrological Atlas (Sevruk, 1985; Kirchhofer, 2000).

## 2.11 Glacier initialisation

Glaciers need decades to centuries to reach an equilibrium under a given stable climate. To take account of these long-term dynamics at the start of the modelling period, the ice cover has to be initialised by the model using a representative quasi-stable climate of this length. This ensures consistency between glacier cover and the driving data, as the interpolated climate data is inherently imprecise compared to the observed glacier cover. Also, the model processes and spatial structure are an imperfect representation of actual conditions, so that observed glacier areas and volumes can not be directly used as initial conditions.

For the proposed model, the glacier area and volume are initialised using a climate period in which the glacier mass balance is known to be close to 0, i.e. in a quasi-equilibrium state (Clarke et al., 2015; Marshall et al., 2011). Since this period is in most cases shorter than the time it takes for a glacier to reach an equilibrium, shorter periods are used successively for 200– 1000 years. Mass balance records around the world have exhibited balanced or even positive budgets in the 1960's until the mid-1970's (Dyurgerov, 2010; Sorg et al., print October 2012; Dyurgerov and Meier, 1997; WGMS and UNEP, 2008).

This is also true for reference glaciers in and close to the case study catchments of this study: The long-term mass balance records of the Griess glacier in the Rhone catchment show a mean of  $-79 \text{ mm a}^{-1}$  between 1962–1980. For the Tien Shan, Dyurgerov (2010) puts forward a regional average mass balance that shows an mean balance of  $-82 \text{ mm a}^{-1}$  between 1960– 1975 (also confirmed by Farinotti et al., 2015). These periods were chosen for the initialisation of the glacier cover in the respective catchments.