# Peer review of "Bridging glacier and river catchment scales: an efficient representation of glacier dynamics in a hydrological model"

_Hydrology and Earth System Sciences, 2016_

## Referee Comment (RC1) · Anonymous Referee #1 · 5 Jul 2016

This manuscript deals with the representation of glaciers in hydrological models. This is a very important issue since, as the authors correctly describe, the link between glacier models and hydrological models is important but often not fully represented in modeling. The manuscript describes a new modeling approach and its application to two catchments. My main concern with the manuscript is that the proposed model (routine) is not fully clear (at least to me) and convincing. Other (minor) concerns relate to the optimization approach and the structure/presentation of the manuscript. Overall, I think this manuscript can make a useful contribution, but requires a major revision (including new computations) to make full justice to the new model development.

There are certain aspects where there are serious doubts whether the chosen ap-

proach is realistic. My doubts might results from misunderstandings, but even in this case this highlights issues with the manuscript. In the end, we all know that a good model fit does not ensure that the model is working for the right reasons. Therefore, it is important to clearly motivate the different equations/approaches being used.

Several of the equations seem to be (semi)empirical, but this is not always clearly stated (e.g. Eqs 2, 3, 14). How generally valid are equations such as Eq 13? This needs to be clearly stated. The annual variation of radiation is a simplified approach of a full geometric estimation, which also would have been possible. While there might be cases where this results in wrong results, Eq 11 might result in general the correct pattern. However, I am a bit confused by the 12 in Eq 12. Sounds like months, but I still do not see why one should divide by 12, sorry.

Eq 16 does not seem to agree with the common view on precip variation with elevation, where precip increases up to a certain elevation and then stays rather constant. Instead Eq 16 results in a symmetric variation below and above some elevation m. Playing around with different values (Tab1) the values also seem unrealistic (factors up to 10, i.e. a precip correction of 1000% and a rather sharp decrease of the correction factors above and below the elevation m (I got a factor of 1.0 for most elevations).

The transformation from snow to ice is not fully clear (P5L15ff). It sounds like all snow is transformed into ice at the end of the summer season (realistic?) but then only if a critical height is exceeded. Sorry, I am lost here: why is snow only transformed into ice if the height is larger than the height at which ice flow would start?

Besides my concerns on the validity of the different equations, the study would also benefit from investigating more clearly the effect of their use. Based on the idea that a model should be as simple as possible, but not simpler, I would suggest to evaluate the contribution/importance of the different equations on the model outcome. This would also allow to better estimate the importance and potential uncertainty effects of the different parts (e.g. debris cover, precip correction, . . .). The way the model is

presented and tested now does not allow this more detailed look 'inside' the model and provides too little motivation on why certain equations were used. For a first paper on a new model I would find a more detailed analysis of its parts highly valuable.

Another question is the effect of the use of units for the glacier between which the ice flow is routed. Based on the description I would expect numerical issues and thus the chosen number of units could have quite some effect, has this been tested?

Parameter optimization and uncertainty: the optimization procedure resulted in different solutions along the Pareto front. While with this approach one does not have to assign weights to the different objective functions, it 1) can result in parameterizations which are very poor according to some criteria and 2) neglects multiple almost similar solutions at one location along the pareto front. I would recommend considering a combined objective function after all for these reasons.

The authors switch partly between past tense and present tense in the description of their work (e.g. p11L18: is and L30 was), please use past tense consistently.

The conclusions read mainly as a summary and do not really summarize the conclusions of this study.

---

## Referee Comment (RC2) · Anonymous Referee #2 · 12 Jul 2016

General assessment

This manuscript introduces a distributed catchment model that incorporates a representation of glacier dynamics. The spatial representation within the model lies midway between semi-distributed models, which represent spatial variability using grouped response units (GRUs), and fully distributed models such as the grid-based DHSVM. The goal is to retain the physical realism attainable through the fully distributed approach while maintaining the computational efficiency of GRU-based models. If successful, such a model would be a valuable tool for making projections of future streamflow variability. Therefore, the topic of the manuscript is highly relevant to the readership of HESS. However, there are a number of points that require attention before

the manuscript could be accepted for publication. In particular, a number of the process representations appear ad-hoc, poorly constrained and/or physically unrealistic. A number of specific comments are provided below.

Specific comments

1. The authors model ice height, which requires an initial estimate of the elevation of the ice bed, which is made using the Glabtop2 approach. How sensitive are the modelled glacier dynamics to uncertainties in the initial elevation estimates?

2. Further to the preceding point, elevations of the glacier hydrotopes would vary through time as the glacier geometry evolves. Is this accounted for in the model – e.g., for air temperature calculation?

3. p. 2 line 34 to p. 3 line 2. Include example reference(s) for greater specificity on this point – perhaps Jost et al. (2012) HESS 16: 849-860.

4. Equation 3 seems ad hoc. Is there an empirical or theoretical basis for it? How sensitive is the model to this specific formulation?

5. Is there any way to validate the avalanche routine? How sensitive are model predictions to leaving it out?

6. Section 2.6. Is the melt factor for glacier ice enhanced relative to the melt factor for snow?

7. Section 2.6. Is the residence time constant? Many empirical and modelling studies have demonstrated a seasonal variation, especially in relation to the timing of snow disappearance.

8. Section 2.6. Glacier outflow is subject to infiltration into a soil layer and surface runoff when that layer saturates. This does not seem realistic. Much, if not most, glacier outflow occurs via subglacial channel networks that evolve through the melt season.

9. Section 2.6. Water is lost from glacier storage by evaporation at a rate determined by the Priestley-Taylor (P-T) equation (note spelling). However, the available energy term in standard applications of the P-T equation would not be appropriate for a glacier. Many express the available energy as Rn – G (Rn = net radiation, G = ground heat flux), which would be better expressed as Rn – M (M = energy consumption by melt) for a glacier. Some applications of the P-T equation leave out the ground heat flux (approximately justified for daily time steps on the basis that the net ground heat flux would be negligible). This approach would also not be appropriate for a glacier. How does the SWIM model represent the P-T equation?

10. Section 2.6. For calculating E using the P-T equation, is the air temperature adjusted to account for conditions within the glacier boundary layer? See papers by Ayala et al. (2015, JGR-Atmos. 3139-3157, DOI: 10.1002/2015JD023137) and references cited therein on the variations of temperature and humidity over a glacier relative to off-glacier measurements.

11. Equations 5 and 14. Are these derivatives or finite differences? If the former, use d_/d_ as the operator; if the latter, use upper-case delta for lack of ambiguity. What numerical scheme is used to solve the equations?

12. Equation 7. "E" has previously been used for evaporation. Use a different symbol.

13. Equation 7. Hydrologists and climatologists commonly use beta for the Bowen ratio. Consider using a different symbol to avoid confusion.

14. Equation 7. Is a temporally and spatially constant sublimation ratio physically realistic? Can the authors draw upon work on sublimation in the dry Andes, for example, to support their parameterization?

15. Equation 7. It seems redundant to compute both evaporation and sublimation at each time step. Evaporation would occur from a melting surface for which a water film covers ice or snow grains. Sublimation would occur from a non-melting surface lacking

a water film.

16. p. 8 line 1. Slope and aspect enhance insolation on equator-facing aspects, not just reduce it.

17. Equation 14. What are the units of C?

18. I have trouble understanding Equation 14. Shouldn't there be lateral flux terms (Q_i in Equation 2) to represent fluxes of sediment from the up-gradient unit and to the down-gradient unit?

19. I may have missed it, but I could not find which years were used for calibration and which for validation. For example, are the time series shown in Figure 4 for the calibration or validation period?

Editorial comments

1. Use the past tense when referring to previous studies.

2. There are a number of minor editorial corrections to be made. Some examples are provided below.

3. p. 7 line 28. Zhao et al. and Winkler et al. are not in the reference list.

4. p. 8 line 7. "sinus" should be "sine"

5. p. 8 line 13. "defuse" should be "diffuse"

6. p. 8 line 7. Use a colon rather than a semi-colon here.

7. p. 11 line 20. Nash-Sutcliffe misspelled

8. p. 11 line 27. . . . at least one objective . . . (?)

9. p. 12 line 28. "complimented" should be "complemented"

---

## Referee Comment (RC3) · Anonymous Referee #3 · 14 Jul 2016

I enjoyed working through this manuscript. The authors have introduced lots of new ideas and sensible approaches to this kind of modeling. I agree that they are addressing an important gap between hydrological and glaciological models, at the catchment scale, and this effort is a genuine bridge between those worlds. There are new ideas and the model is applied in two interesting and well-selected locations. The paper is generally well-written, with strong conclusions that are well-supported by the model results. I think this work will find a receptive audience and it is likely that others will build on the model presented here. For these reasons, I recommend publication, with a number of minor points and perhaps one or two larger concerns to be addressed.

Main concerns

1. I have one specific concern about the model, which might just need more explanation or might suggest a 'push-back' towards more substantial revisions. On p.5, l.17 and Eq. (1), the critical snow thickness to be ice does not make sense to me. Hc. For typical slopes, like 10 degrees, tan\alpha $\sim$ 0.1 and Hc = 100 m. Does this mean that snow that survives the summer does not turn into ice unless there is at least 100 m of snow/ice accumulated? What about the snow thermal and albedo properties, etc., those should match ice after one year. Also, ice that is thinner than this creeps and slides. It does not wait until it reaches a critical shear stress before it starts moving. That is a misinterpretation of \tau_s, which is more a 'balance' value where steady-state fluxes allow an equilibrium ice thickness. If I understand correctly, this seems odd an arbitrary for a glacier not to exist, thermodynamically and mechanically, until this much ice has accumulated. It would preclude many of today's present glaciers (and parts of them), which are thinner than this.

I also did not understand Eq. 3 or the area treatment on the next page, this could be explained more clearly I think. As I read it, Hc is maintained while area decreases in the lowest elevation band? I like this general idea, it is a nice new idea, but it seems unrealistic to maintain a steep and non-thinning layer of glacier ice while the area retreats. Shouldn't both decline at once, following a realistic volume-area relationship or what one would expect for a 'wedge-shaped' terminus?

2. As the title of my review suggests, I did struggle with whether this model is sufficiently physically-based and state-of-the-art to actually be useful. That sounds harsh, perhaps, but there are more complex and realistic models out there (many of them cited by the authors), and the model proposed here has several free parameters of an empirical nature, e.g., degree day factors, which are not actual physical variables. These are tuned to observed discharge in the specific basins, and results are reasonable, but how portable are they in space and time? The authors do nicely balance complexity with pragmatism, with a relatively simple treatment of a lot of the processes, but perhaps appropriate for the large-scale objectives. I do agree with the authors that forcing

data are not commonly available for more complex models (e.g., energy balance melt models, or more detailed glacier processes such as sliding). I just have concerns that this model is heavily parameterized and tuned in ways that are not representing the actual physics, which make it unclear how broadly useful it is. For instance, no attention is paid to conservation of energy or mass at the catchment scale; precipitation and mass balance are scaled as necessary, and melt rates are turned up or down ad hoc in order to match discharge observations. In the final lines, the authors note an intention to use this modeling approach in future projections, but is it reasonable to take climate model precipitation and energy fields and manipulate them in this way, paying no heed to basic conservation? I have concerns that this is too far from constrained reality. Still, this model presents an initial step into coupled glacio-hydrological modeling that has yet to be done well at large scales. For this reason I don't think my concern here is fatal.

Some specific questions on this point:

How do the final parameter sets vary for the different catchments/sub-catchments? Are there generally sensible, robust, and repeatable parameter values, that you would feel comfortable to use in other environments or in future projections?

Is the glacier melt model the same as that for the SWIM snow model? I did not realize that degree day melt models were still in broad use, scientifically. The range of melt factors explored here (Table 1) looks like values that are common literature values used for snow and ice melt, without incorporating the effects of shading, aspect, debris cover, etc. I have trouble to imagine that this fully represents the range of potential values.

Daily mean temperature is also simplistic when it comes to estimating PDD for snow and ice melt. e.g., a mean temperature of -1 C means no melt, although much of the day will be above 0 C. Minimum and maximum temperature are widely available and can be used to generate a daily temperature cycle - can this be considered?

In general the modeling approach emphasizes the model parsimony, that it does not

need many input variables, and only things like temperature that are 'more known'. But it does need regional mass balances, across individual glaciers: both in balance, for the spinup, and maybe also in time, for the model calibration - I was less sure here. Are these widely known, or known well enough to be able to use this model in much of the world? It seems like this is 'higher order' knowledge then some of the basic meteorological variables you would need for an energy balance model.

Back to degree day factors.. I don't understand why these are based on length of day. Why not actually calculate potential shortwave radiation as a function of latitude, day of year, slope, aspect and shading? This is a more direct and realistic way to include this effect, and can be pre-processed easily. Do the length of day calculations include slope and aspect effects, and shading?

And some minor points:

p.1, l.20, "strongly heterogeneous processes like glacier dynamics" - is that really true? glacier dynamics work roughly the same way everywhere. But ice thickness and slope vary strongly in space.

p.3, l.30, "is described"

p.5, l.3, there is some jargon throughout, like 'cleaning area' - please define. Also 'hydrotope', referenced below on this page. How does a hydrotope compare with an HRU, or are they equivalent?

p.5, l.12, 30 cm of soil cover actually seems like a lot for steep alpine terrain, which is more likely bare rock. Am I mid-understanding here?

p.7, Eq. 10. I like this in general, the approach to separate melt and sublimation. But note that in fact M in field-based PDD calibration studies includes sublimation, i.e. it is actually observed ablation, M+S. But generally S « M where PDD factors are being calibrated, so this is maybe OK. This is not true everywhere though.

p.9, debris discussion. I was not sure that these are the main processes involved. Also

consider landslide/rock avalanche debris? Also aerosol deposition, which can be a blanket or it can also be concentrated by ablation and streams, e.g. where streams intersect crevasses. These processes should at least be noted, as they can be locally or regionally important.

p.12, l.28, "complemented". Next line down, "its"

p.13, l.19, implemented

Figure 4, outlet of the Rhone. It looks like there are diurnal cycles here - is this really daily discharge, or is it hourly? If the former, what are the oscillations?

p.15, l.13, comparison "is" shown

p.16, l.8, principal

p.16, l.9, I don't think one year can have a climate. The year's weather?

p.16, l.14, "varies"

p.18, l.16, I don't think it is wise to say that it covers "all major glacier processes". As the authors point out, things like glacier sliding, surging, and calving are not included, and these could be considered major processes. Also seasonal albedo evolution and many other details of glacier ablation.

---

## Author Comment (AC1) · 7 Oct 2016

Thank you for the review and constructive comments. We have responded point by point below with your comments in italic.

*This manuscript deals with the representation of glaciers in hydrological models. This is a very important issue since, as the authors correctly describe, the link between glacier models and hydrological models is important but often not fully represented in modeling. The manuscript describes a new modeling approach and its application to two catchments. My main concern with the manuscript is that the proposed model (routine) is not fully clear (at least to me) and convincing. Other (minor) concerns relate to the optimization approach and the structure/presentation of the manuscript. Overall,*

*I think this manuscript can make a useful contribution, but requires a major revision (including new computations) to make full justice to the new model development.*

*There are certain aspects where there are serious doubts whether the chosen approach is realistic. My doubts might results from misunderstandings, but even in this case this highlights issues with the manuscript. In the end, we all know that a good model fit does not ensure that the model is working for the right reasons. Therefore, it is important to clearly motivate the different equations/approaches being used.*

Thank you for this assessment and the suggestions for improvement. We have revised the manuscript to make the model description more clear and convincing, emphasizing the proven and tested nature of most of the equations (specifically in sections 2.3, 2.4, 2.7, 2.9). The description of model calibration using a multi-objective evolutionary optimisation in section 2.12 is extended. To allow a deeper look 'inside' the model, we have additionally provided a sensitivity analysis of all calibration parameters with regard to the calibration objectives in the supplementary material.

*Several of the equations seem to be (semi)empirical, but this is not always clearly stated (e.g. Eqs 2, 3, 14). How generally valid are equations such as Eq 13? This needs to be clearly stated. The annual variation of radiation is a simplified approach of a full geometric estimation, which also would have been possible. While there might be cases where this results in wrong results, Eq 11 might result in general the correct pattern. However, I am a bit confused by the 12 in Eq 12. Sounds like months, but I still do not see why one should divide by 12, sorry.*

The empirical or semi-empirical nature of the equations was further described as appropriate. Equation 3 was improved as suggested by reviewer #3. The parametrization of Eq 13 is now better described in section 2.9. As you correctly observe, a full geometric estimation of potential radiation is possible but it would introduce an additional calibration parameter, which we aimed to avoid. The 12 in Eq. 12 are the potential hours of sunlight at the equinox to scale the potential hours for each unit. It is added to

section 2.8: (12 signifying the potential hours at the equinox on an unshaded horizontal surface).

*Eq 16 does not seem to agree with the common view on precip variation with elevation, where precip increases up to a certain elevation and then stays rather constant. Instead Eq 16 results in a symmetric variation below and above some elevation m. Playing around with different values (Tab1) the values also seem unrealistic (factors up to 10, i.e. a precip correction of 1000% and a rather sharp decrease of the correction factors above and below the elevation m (I got a factor of 1.0 for most elevations).*

Eq. 16 was only introduced to overcome the precipitation discrepancies of the data-scarce Upper Aksu catchment (only one station with long-term high-altitude observations is located west of the catchment), and it is based on the approach by Immerzeel et al. (2012, 2015). We have made this more clear in section 2.10, paragraph 2. The maximal correction factor of 10 was corrected to upper boundaries applicable to only the Aksu catchment (6) in accordance with Aizen (1995), who report precipitation totals of 1000mm/a for the catchment at high altitudes. However, for the Upper Rhone this precipitation correction function was not necessary, and correction factors from the Swiss Hydrological Atlas (Sevruk, 1985; Kirchhofer, 2000) were used instead.

*The transformation from snow to ice is not fully clear (P5L15ff). It sounds like all snow is transformed into ice at the end of the summer season (realistic?) but then only if a critical height is exceeded. Sorry, I am lost here: why is snow only transformed into ice if the height is larger than the height at which ice flow would start?*

In the model, we assume that the snow pack is turned into glacial ice if the snowpack exceeds the critical height. All snow processes are governed by the snow module of the SWIM model (based on the snow module by Gelfan et al., 2004), which describes the share of ice and water in the snowpack as well. The critical height is used to determine at what point the snowpack is subject to creep and slip. The global shear stress (taken from Cuffey and Paterson (2010)) is the best estimate we can find for

data scarce catchments. This is now clarified in section 2.3.

*Besides my concerns on the validity of the different equations, the study would also benefit from investigating more clearly the effect of their use. Based on the idea that a model should be as simple as possible, but not simpler, I would suggest to evaluate the contribution/importance of the different equations on the model outcome. This would also allow to better estimate the importance and potential uncertainty effects of the different parts (e.g. debris cover, precip correction,...). The way the model is presented and tested now does not allow this more detailed look 'inside' the model and provides too little motivation on why certain equations were used. For a first paper on a new model I would find a more detailed analysis of its parts highly valuable.*

Thanks for this suggestion! To allow a closer look into the effects of the different parameters, we have provided a sensitivity analysis for all calibration parameters (based on their correlation with the calibration objectives) as well as an analysis of specific results produced with different parameter sets. The latter includes an analysis of the ice flow rheology parameter and avalanching. The sensitivity analysis and its results are presented in the supplementary material.

*Another question is the effect of the use of units for the glacier between which the ice flow is routed. Based on the description I would expect numerical issues and thus the chosen number of units could have quite some effect, has this been tested?*

If by numerical issues the exchange of ice between irregular-sized units is meant, we prevented these issues by aggregating ('cleaning') the glacier units to a minimum size to ensure that similarly sized units are produced. The elevation intervals in valleys and on hillslopes are chosen with the unit size distribution in mind. This is now better explained in section 2.2 and min., mean and max. unit sizes for both catchments are given.

*Parameter optimization and uncertainty: the optimization procedure resulted in different solutions along the Pareto front. While with this approach one does not have to*

*assign weights to the different objective functions, it 1) can result in parameterizations which are very poor according to some criteria and 2) neglects multiple almost similar solutions at one location along the pareto front. I would recommend considering a combined objective function after all for these reasons.*

We agree that there are drawbacks in using a multi-objective calibration. However, there are advantages and drawbacks in both methods. The algorithm used here and variants thereof have been successfully used in data-scarce, glacierised catchments (e.g. Duethmann et al., 2014, 2015; Rahman et al., 2012). It is used here as a tool to find an efficient optimisation of multiple objective functions without assigning weights prior to calibration. A single combined objective function would require weighing up an acceptable error in river discharge with an acceptable error in mass balance before calibration. Since they are not known and may vary between catchments, we feel that a multi-objective approach is more appropriate here. The possibility of very poor results by some criteria is eliminated by choosing a subset of parameter sets with a minimal performance criteria. The min., median and max. values of the results are given in table 4. We have now added the minimal criteria (that were previously only briefly mentioned in the results) to the calibration section 2.12.

*The authors switch partly between past tense and present tense in the description of their work (e.g. p11L18: is and L30 was), please use past tense consistently.*

Thank you, that is rectified.

*The conclusions read mainly as a summary and do not really summarize the conclusions of this study.*

We feel that the extracted concluding sentences listed below are indeed summarising the conclusions of the manuscript (and referee 3 seems to agree: 'The paper is generally well-written, with strong conclusions that are well-supported by the model results.'):

- The new approach to representing individual glaciers and their ice dynamics in a hydrological model bridges the gap between distributed, physically based glacier dynamics models – that are typically only applicable to single glaciers or small glacier groups – and large-scale empirical glacio-hydrological models.

- This allows for accurate and integrated glaciological and hydrological assessments of entire, highly glacierised catchments.

- The intermediate complexity enables ensemble modelling approaches for calibration and scenario analysis by radically reducing computing time compared to fully distributed glacier models.

- The calibration yielded good results compared to both discharge and glaciological observations, but performance depends on data quality – precipitation observations in particular.

- The parameter uncertainty is comparable to uncertainties of glaciological observations (e.g. glaciological or geodetic area and mass balance observations) but may become large over longer simulation periods due to the variable initialisations.

- In data-scarce catchments, the model highlights the need for precipitation correction and is able to inform the method of correction by initialising ice cover and calibrating the model using discharge, glacier distribution and glacier mass balance in the multi-objective calibration procedure.

- The model helps to prevent overestimations of glacier melt in-lieu for negative biases in precipitation observations that are ubiquitous in mountainous catchments.

- The application to the arid Upper Aksu catchment shows that adequately simulating glacier dynamics (including accurate rates of accumulation and ablation) is

vital to properly model this and similar river basins due to their high contribution of glacier melt to discharge.

- The intermediate complexity of the developed glacio-hydrological model means that the model is well adapted to large, partially glacierised and data-scarce catchments, as they are often found in High Asia and other mountain ranges of the world.

Please also note the supplement to this comment:
http://www.hydrol-earth-syst-sci-discuss.net/hess-2016-272/hess-2016-272-AC1-supplement.pdf

---

## Author Comment (AC2) · 7 Oct 2016

Thank you for the review and constructive comments. We have responded point by point below with your comments in italic.

*General assessment: This manuscript introduces a distributed catchment model that incorporates a representation of glacier dynamics. The spatial representation within the model lies midway between semi-distributed models, which represent spatial variability using grouped response units (GRUs), and fully distributed models such as the grid-based DHSVM. The goal is to retain the physical realism attainable through the fully distributed approach while maintaining the computational efficiency of GRU-based models. If successful, such a model would be a valuable tool for making projections of*

*future streamflow variability. Therefore, the topic of the manuscript is highly relevant to the readership of HESS. However, there are a number of points that require attention before the manuscript could be accepted for publication. In particular, a number of the process representations appear ad-hoc, poorly constrained and/or physically unrealistic.*

Thank you for both criticism and praise. We have tried to clarify the process descriptions and emphasise the need for relatively simple but robust formulations in the face of data scarcity. Parsimonious process descriptions (unfortunately) often rely on empirical parameters that must be chosen wisely based on available literature sources. This was our choice for the glacio-hydrological model in hand, which tries to balance physical process descriptions and data availability. In addition, we have illustrated the sensitivity of the individual parameters included in the glacier module in a sensitivity analysis (included in supplementary material). Answers to specific comments are given below.

*1. The authors model ice height, which requires an initial estimate of the elevation of the ice bed, which is made using the Glabtop2 approach. How sensitive are the modelled glacier dynamics to uncertainties in the initial elevation estimates?*

The model does not need an initial estimate of the ice bed, instead we are using the Glabtop2 simulations to calibrate ice thickness via the rheology term $\chi$ in Eq. 2. The average error of Glabtop2 was shown to be 7% in larger glacierised regions (Frey et al., 2014), which is negligible given the uncertainties of the catchment-wide thickness calibration using a global shear stress in our model.

*2. Further to the preceding point, elevations of the glacier hydrotopes would vary through time as the glacier geometry evolves. Is this accounted for in the model – e.g., for air temperature calculation?*

No this is not accounted for because the glacier thickness will not decrease below the critical height. The glacier area is reduced when the unit melts further as illustrated in

[Figure]

Fig. 2. While accounting for it would be possible, it would only have a limited impact for the above reason. Also, the model set up would rely on the externally modelled thickness, e.g. from Glabtop2, which is both error-prone and may not always be available.

*3. p. 2 line 34 to p. 3 line 2. Include example reference(s) for greater specificity on this point – perhaps Jost et al. (2012) HESS 16: 849-860.*

Thank you, we included it.

*4. Equation 3 seems ad hoc. Is there an empirical or theoretical basis for it? How sensitive is the model to this specific formulation?*

Since the shape of the glacier units is unknown, the initial formulation assumes that they are rectangular blocks where the width is approximated by the squareroot of the area. Referee #3 has suggested the slightly more realistic formulation of a wedge-shaped front, relying on the same assumption for the width. It is now included in section 2.4.

*5. Is there any way to validate the avalanche routine? How sensitive are model predictions to leaving it out?*

Validation is difficult but observed glacier outlines above the ELA give an indication of critical slope angles, but steep, high elevation terrain is also prone to misclassification in glacier outlines. Leaving it out would prevent adjusting the glacier hypsometry above the ELA to observations and also reduce snow accumulation.

*6. Section 2.6. Is the melt factor for glacier ice enhanced relative to the melt factor for snow?*

No, the melt factors for snow and ice are calibrated separately but the snow melt factor is also scaled by the sun hours to account for aspect and terrain shading.

*7. Section 2.6. Is the residence time constant? Many empirical and modelling studies have demonstrated a seasonal variation, especially in relation to the timing of snow*

[Figure]

*disappearance.*

The residence time is constant mainly to avoid parameter redundancy of a parameter that is not very sensitive at the catchment scale and the daily time step (this has been checked). It is, however, included as a single parameter to account for the delays between melt and runoff, where it is important, e.g. in smaller catchments.

*8. Section 2.6. Glacier outflow is subject to infiltration into a soil layer and surface runoff when that layer saturates. This does not seem realistic. Much, if not most, glacier outflow occurs via subglacial channel networks that evolve through the melt season.*

Thanks! Yes, this is correct and most discharge is generated by 'surface runoff' because the thin (30-100cm) subglacial soil/sediment layer is saturated throughout the melting season. However, the process of evolving channel networks is not included as it has little influence on discharge at the catchment scale.

*9. Section 2.6. Water is lost from glacier storage by evaporation at a rate determined by the Priestley-Taylor (P-T) equation (note spelling). However, the available energy term in standard applications of the P-T equation would not be appropriate for a glacier. Many express the available energy as Rn-G (Rn = net radiation, G = ground heat flux), which would be better expressed as Rn-M (M = energy consumption by melt) for a glacier. Some applications of the P-T equation leave out the ground heat flux (approximately justified for daily time steps on the basis that the net ground heat flux would be negligible). This approach would also not be appropriate for a glacier. How does the SWIM model represent the P-T equation?*

The P-T equation is the standard method of SWIM to estimate potential evapotranspiration, which was adopted for lack of appropriate equations for ice surfaces without solving the full energy balance at the glacier surface. We instead assume that melt water left in the linear reservoir is available for free surface evaporation (water saturated firn, supraglacial puddles, ponds and lakes), which is reasonable given the data
constraints.

*10. Section 2.6. For calculating E using the P-T equation, is the air temperature adjusted to account for conditions within the glacier boundary layer? See papers by Ayala et al. (2015, JGR-Atmos. 3139-3157, DOI: 10.1002/2015JD023137) and references cited therein on the variations of temperature and humidity over a glacier relative to off-glacier measurements.*

Given the scope of the model for data-scarce catchments, this process is difficult to represent at the large catchment scale with gridded temperature data. We also do not consider glacier flow lines because they are mostly only available for large valley glaciers.

*11. Equations 5 and 14. Are these derivatives or finite differences? If the former, use d_/d_ as the operator; if the latter, use upper-case delta for lack of ambiguity. What numerical scheme is used to solve the equations?*

These are simple first order difference equations, we corrected the equations accordingly with capital delta over time step d (day).

*12. Equation 7. "E" has previously been used for evaporation. Use a different symbol.*

Thank you, evaporation is renamed to EP.

*13. Equation 7. Hydrologists and climatologists commonly use beta for the Bowen ratio. Consider using a different symbol to avoid confusion.*

Thank you, we changed it to $\Gamma$.

*14. Equation 7. Is a temporally and spatially constant sublimation ratio physically realistic? Can the authors draw upon work on sublimation in the dry Andes, for example, to support their parameterization?*

See below answer on questions 14 and 15.
*15. Equation 7. It seems redundant to compute both evaporation and sublimation at each time step. Evaporation would occur from a melting surface for which a water film covers ice or snow grains. Sublimation would occur from a non-melting surface lacking a water film.*

Thanks for these comments! While sublimation ratios most definitely vary significantly over a catchment area and time, the factors (shortwave radiation, latent heat flux, wind speed, roughness etc. as found on low-latitude glaciers) to determine time and space varying sublimation are not readily available at the catchment scale. We thus rely on the findings of point observations (eg. Winkler et al. 2009; Zhang et al., 2012; MÃÂűlg et al. 2009) and theoretical considerations to at least include sublimation as a first order estimation over the entire catchment and considering annual mass balances only. In our opinion, this is justified given the relative importance of sublimation.

*16. p. 8 line 1. Slope and aspect enhance insolation on equator-facing aspects, not just reduce it.*

Thanks, we corrected to 'alter' instead. The enhancing effect is considered through sunshine hours greater than 12 on a given day.

*17. Equation 14. What are the units of C?*

C is the concentration or fraction of debris in the ice column and is thus dimensionless (added to section 2.9).

*18. I have trouble understanding Equation 14. Shouldn't there be lateral flux terms ($Q_i$ in Equation 2) to represent fluxes of sediment from the up-gradient unit and to the down-gradient unit?*

We have corrected the equation to be a valid first order difference equation describing the change in concentration over one day (as in the model code). We changed melt to ablation to also consider sublimation. The equation is now better described in section 2.9: The first term changes the concentration according to the ratio of the specific

mass balance $(A - H_s)$. The second term describes the 'dilution' of the ice flux from the upstream unit $(C_u - C_g)$.

*19. I may have missed it, but I could not find which years were used for calibration and which for validation. For example, are the time series shown in Figure 4 for the calibration or validation period?*

Thanks for pointing that out, we have added these periods to section 2.12 and table 3. For the Rhone catchment: calibration period 1980–1995, validation period 1996–2010. For the Aksu catchment: calibration period 1971–1978/1982 (Xiehela/Sary Djaz), validation period 1978/1982–1987/1996.

*Editorial comments 1. Use the past tense when referring to previous studies. 2. There are a number of minor editorial corrections to be made. Some examples are provided below. 3. p. 7 line 28. Zhao et al. and Winkler et al. are not in the reference list. 4. p. 8 line 7. "sinus" should be "sine" 5. p. 8 line 13. "defuse" should be "diffuse" 6. p. 8 line 7. Use a colon rather than a semi-colon here. 7. p. 11 line 20. Nash-Sutcliffe misspelled 8. p. 11 line 27. ...at least one objective...(?) 9. p. 12 line 28. "complimented" should be "complemented"*

Thank you for pointing out these editorial changes, they have all been integrated.

Please also note the supplement to this comment:
http://www.hydrol-earth-syst-sci-discuss.net/hess-2016-272/hess-2016-272-AC2-supplement.pdf
* * *
[Figure]

**Supplement:**

**Supplementary material to "Bridging glacier and river catchment scales: an efficient representation of glacier dynamics in a hydrological model" by Wortmann et al.**

October 7, 2016

**1   Parameter sensitivity**

[Figure]

Figure 1: Calibration parameter sensitivity with regard to the four objective functions: Nash-Sutcliffe Efficiency (NSE), percentage bias in the water balance (PB), RMSE of the initialised glacier area hypsometry (A) and the RMSE of annual mass balance (MB). The parameters are with further details given in Table 1: $\delta_s$ (Snow Degree-Day factor), $T_s$, $T_m$ (Snow fall and melt threshold temperatures), $t_e$ (Temperature lapse rate), $\delta_g$ (Ice Degree-Day factor), $b_r$ (Residual mass balance during initialisation), $c$ and $a$ (Maximum precipitation correction factor and maximum precipitation gradient, both only applied to the Aksu catchment), $E_c$ (evaporation correction factor), $R_2$, $R_4$ (routing coefficients) and $S_c$ (saturated conductivity correction). The partial correlation coefficient of the parameters over the 5000 calibration runs were averaged over all catchments. As the parameters $E_c$, $R_2$, $R_4$ and $S_c$ have no impact on A and MB, the coefficients were excluded here.

**2  Calibrated parameter values**

Table 1: Calibrated parameters (min., median, max.)  for both investigated catchments over the best 25 parameter sets. Refer to Table 1 for a description of the parameters.

| Parameter | Upper Aksu | | | Upper Rhone | | |
|---|---|---|---|---|---|---|
| | min | median | max | min | median | max |
| $\delta_s$ | 0.32 | 0.38 | 0.41 | 0.27 | 0.40 | 0.50 |
| $T_s$ | 2.2 | 3.3 | 3.8 | -2.3 | 1.3 | 1.9 |
| $T_m$ | -2.0 | -1.2 | -0.7 | -1.2 | -0.5 | 1.6 |
| $t_e$ | -0.78 | -0.72 | -0.68 | -0.60 | -0.56 | -0.49 |
| $\delta_g$ | 5.8 | 8.6 | 11.6 | 6.1 | 8.8 | 10.3 |
| $b_r$ | -281 | -250 | -182 | -129 | -79 | 149 |
| $c^*$ | 3.1 | 3.6 | 3.9 | | | |
| $a^*$ | 0.31 | 0.34 | 0.37 | | | |
| $E_c$ | 0.61 | 0.79 | 1.15 | 0.74 | 1.08 | 1.41 |
| $R_2$ | 0.5 | 1.1 | 3.1 | 1.3 | 3.8 | 5.0 |
| $R_4$ | 0.9 | 2.7 | 4.1 | 1.6 | 4.6 | 5.0 |
| $S_c$ | 0.6 | 1.2 | 1.9 | 1.0 | 1.3 | 1.7 |

*only used for the data-scarce Aksu catchment

---

## Author Comment (AC3) · 7 Oct 2016

Thank you for the review and constructive comments. We have responded point by point below with your comments in italic.

*I enjoyed working through this manuscript. The authors have introduced lots of new ideas and sensible approaches to this kind of modeling. I agree that they are addressing an important gap between hydrological and glaciological models, at the catchment scale, and this effort is a genuine bridge between those worlds. There are new ideas and the model is applied in two interesting and well-selected locations. The paper is generally well-written, with strong conclusions that are well-supported by the model results. I think this work will find a receptive audience and it is likely that others will*

[Figure]

*build on the model presented here. For these reasons, I recommend publication, with a number of minor points and perhaps one or two larger concerns to be addressed.*

*Main concerns:*

*1. I have one specific concern about the model, which might just need more explanation or might suggest a 'push-back' towards more substantial revisions. On p.5, l.17 and Eq. (1), the critical snow thickness to be ice does not make sense to me. Hc. For typical slopes, like 10 degrees, tan n alpha 0.1 and Hc = 100 m. Does this mean that snow that survives the summer does not turn into ice unless there is at least 100 m of snow/ice accumulated? What about the snow thermal and albedo properties, etc., those should match ice after one year. Also, ice that is thinner than this creeps and slides. It does not wait until it reaches a critical shear stress before it starts moving. That is a misinterpretation of $\tau_s$, which is more a 'balance' value where steady-state fluxes allow an equilibrium ice thickness. If I understand correctly, this seems odd an arbitrary for a glacier not to exist, thermodynamically and mechanically, until this much ice has accumulated. It would preclude many of today's present glaciers (and parts of them), which are thinner than this.*

In the model we assume that the snow pack is turned into glacial ice if the snowpack exceeds the critical height. All snow processes are governed by the snow module of SWIM (based on the snow module by Gelfan et al., 2004), which describes the share of ice and water in the snowpack as well. The critical height is used to determine at what point the snowpack is subject to creep and basal slip according to Cuffey and Paterson (2010) and Marshall et al. (2012). While shear stress certainly deviates from the global mean of $10^5$Pa, it is the best estimate in catchments without better estimates.

*I also did not understand Eq. 3 or the area treatment on the next page, this could be explained more clearly I think. As I read it, Hc is maintained while area decreases in the lowest elevation band? I like this general idea, it is a nice new idea, but it seems unrealistic to maintain a steep and non-thinning layer of glacier ice while the area re-*

*treats. Shouldn't both decline at once, following a realistic volume-area relationship or
what one would expect for a 'wedge-shaped' terminus?*

Thanks! We have taken on your idea of a wedge-shaped front rather than a steep one
as initially used. The length $l$ of the wedge is proportional to the glacier thickness of
the unit, and the melt area is proportional to the hypotenuse of the wedge (section 2.4
was extended). As before, the width is assumed to be the square-root of the unit area.

$$l = \sqrt{A_u} \cdot \frac{H}{H_c} \tag{1}$$

The melt area $A_m$ $[km^2]$ is thus:

$$A_m = \sqrt{A_u} \cdot \sqrt{l^2 + H_c^2} \tag{2}$$

*2. As the title of my review suggests, I did struggle with whether this model is suffi-
ciently physically-based and state-of-the-art to actually be useful. That sounds harsh,
perhaps, but there are more complex and realistic models out there (many of them cited
by the authors), and the model proposed here has several free parameters of an em-
pirical nature, e.g., degree day factors, which are not actual physical variables. These
are tuned to observed discharge in the specific basins, and results are reasonable, but
how portable are they in space and time? The authors do nicely balance complexity
with pragmatism, with a relatively simple treatment of a lot of the processes, but per-
haps appropriate for the large-scale objectives. I do agree with the authors that forcing
data are not commonly available for more complex models (e.g., energy balance melt
models, or more detailed glacier processes such as sliding). I just have concerns that
this model is heavily parameterized and tuned in ways that are not representing the
actual physics, which make it unclear how broadly useful it is. For instance, no atten-
tion is paid to conservation of energy or mass at the catchment scale; precipitation and
mass balance are scaled as necessary, and melt rates are turned up or down ad hoc in
order to match discharge observations. In the final lines, the authors note an intention*

*to use this modeling approach in future projections, but is it reasonable to take climate model precipitation and energy fields and manipulate them in this way, paying no heed to basic conservation? I have concerns that this is too far from constrained reality. Still, this model presents an initial step into coupled glacio-hydrological modeling that has yet to be done well at large scales. For this reason I don't think my concern here is fatal.*

We understand the concern that the process descriptions may not be state-of-the-art and rely on several free and empirical parameters. The model's scope, however, are catchments with only the most basic observations used for calibration (discharge, glacier outlines) and very limited, error-prone driving data (temperature, precipitation), as they are found in abundance in High Asia. The sampling density of precipitation is of particular concern; many catchments don't even have a single gauge station. More physically based approaches (which may be described as state-of-the-art) necessarily rely on better driving data and have therefore only been successfully applied to small domains. One can only close the catchment-wide mass and energy balance if their input is known, which is decidedly not the case here (precipitation is mostly vastly underestimated, incoming shortwave radiation is extrapolated over several 100km at best). Our glacier-hydrology integrated approach relies therefore on discharge, glacier extent and sensible catchment-wide, annual mass (liquid and solid) balance values (snow accumulation, sublimation, evapotranspiration, groundwater yield). We have added these water balance values to the results section 4.4 to support the validity of the parametrisation.

*Some specific questions on this point: How do the final parameter sets vary for the different catchments/sub-catchments? Are there generally sensible, robust, and repeatable parameter values, that you would feel comfortable to use in other environments or in future projections?*

The parameter ranges are selected to the best available knowledge of the respective catchment ensuring that both water and ice balance terms are reasonable. We have

listed the resultant min., median and max. values of our best 25 parameter sets for each catchment in the supplementary material. The parameter ranges are comparable but are of course always subject to calibration when implemented in a different catchment.

*Is the glacier melt model the same as that for the SWIM snow model? I did not realize that degree day melt models were still in broad use, scientifically. The range of melt factors explored here (Table 1) looks like values that are common literature values used for snow and ice melt, without incorporating the effects of shading, aspect, debris cover, etc. I have trouble to imagine that this fully represents the range of potential values.*

SWIM has a separate snow model (based on Gelfan et al., 2004) that also uses a degree-day approach but is also able to account for ice and water content as well as refreezing. We have attempted to alter (allowing both increase and decrease) the melt factors to commonly known effects such as terrain shading and debris cover in our model, see in sections 2.8–2.9.

*Daily mean temperature is also simplistic when it comes to estimating PDD for snow and ice melt. e.g., a mean temperature of -1 C means no melt, although much of the day will be above 0 C. Minimum and maximum temperature are widely available and can be used to generate a daily temperature cycle - can this be considered?*

This effect is accounted for by the melt temperature threshold that is commonly used in PDD models which also accounts for variations of the melt temperature due to latent heat fluxes (Hock, 2003). The model time step is daily, i.e. a diurnal temperature cycle cannot be accounted for.

*In general the modeling approach emphasizes the model parsimony, that it does not need many input variables, and only things like temperature that are 'more known'. But it does need regional mass balances, across individual glaciers: both in balance, for the spinup, and maybe also in time, for the model calibration - I was less sure here. Are these widely known, or known well enough to be able to use this model in much*

*of the world? It seems like this is 'higher order' knowledge then some of the basic meteorological variables you would need for an energy balance model.*

We agree that the regional mass balances are indeed a higher order knowledge of a catchment, but they only need be known for longer periods and not at the same time step as the model runs at. The emergence of satellite based geodetic mass balance studies has increased the coverage of such data dramatically over the past decade or so and the model can make good use of such new data. A possible alternatives are reference glaciers that have annual mass balance records. The World Glacier Monitoring Service (WGMS) has a rich database for both remote sensing based and in-situ observations.

*Back to degree day factors.. I don't understand why these are based on length of day. Why not actually calculate potential shortwave radiation as a function of latitude, day of year, slope, aspect and shading? This is a more direct and realistic way to include this effect, and can be pre-processed easily. Do the length of day calculations include slope and aspect effects, and shading?*

Potential shortwave radiation is indeed an alternative to length of day (the terrain analysis algorithm used here in GRASS actually outputs this as well), yet it does require another empirical parameter to scale the radiation to melt factors while being highly correlated with potential hours of sun. We aimed to avoid this by sticking to the proven positive degree-day approach but scaling it by the day length or potential duration of sunshine the glacier unit receives also taking account of aspect and shading effects.

*And some minor points:*

Thank you for pointing out those minor points to improve understanding. They were all integrated and some answered in more detail below.

*p.1, l.20, "strongly heterogeneous processes like glacier dynamics" - is that really true? glacier dynamics work roughly the same way everywhere. But ice thickness and slope*

[Figure]

*vary strongly in space.*

This was changed to 'spatially strongly heterogeneous water balance components'.

*p.3, l.30, "is described"*

*p.5, l.3, there is some jargon throughout, like 'cleaning area' - please define. Also 'hydrotope', referenced below on this page. How does a hydrotope compare with an HRU, or are they equivalent?*

Cleaning was replaced by aggregation and yes, hydrotope and HRU are equivalent, which has been indicated in this sentence.

*p.5, l.12, 30 cm of soil cover actually seems like a lot for steep alpine terrain, which is more likely bare rock. Am I mid-understanding here?*

Added: All hillslopes are given the land cover category bare soil, i.e. the soil cover is treated as loosely consolidated, unvegetated soil that describes the loose gravel and small fluvial vans.

*p.7, Eq. 10. I like this in general, the approach to separate melt and sublimation. But note that in fact M in field-based PDD calibration studies includes sublimation, i.e. it is actually observed ablation, M+S. But generally S « M where PDD factors are being calibrated, so this is maybe OK. This is not true everywhere though.*

Added: Degree-Day factors observed by ablation stakes include sublimation. They should therefore be compared to simulated ablation and not used for the ice melt term without calibration.

*p.9, debris discussion. I was not sure that these are the main processes involved. Also consider landslide/rock avalanche debris? Also aerosol deposition, which can be a blanket or it can also be concentrated by ablation and streams, e.g. where streams intersect crevasses. These processes should at least be noted, as they can be locally or regionally important.*

Thanks! These processes were further mentioned in Methods/Debris section as well as the Discussion.

*p.12, l.28, "complemented". Next line down, "its" Done*

*p.13, l.19, implemented*

*Figure 4, outlet of the Rhone. It looks like there are diurnal cycles here - is this really daily discharge, or is it hourly? If the former, what are the oscillations?*

Added to Figure 4 caption: 'The oscillations in the Port du Scex discharge are the effects of Sunday dam closures.' As mentioned in the catchment description (section 3.2), they Sundays were excluded from the observations to reduce the effect on calibration results.

*p.15, l.13, comparison "is" shown*

*p.16, l.8, principal*

*p.16, l.9, I don't think one year can have a climate. The year's weather?*

*p.16, l.14, "varies"*

*p.18, l.16, I don't think it is wise to say that it covers "all major glacier processes". As the authors point out, things like glacier sliding, surging, and calving are not included, and these could be considered major processes. Also seasonal albedo evolution and many other details of glacier ablation.*

Changed to: It covers most glacier processes relevant to simulating catchment discharge including [...]

Please also note the supplement to this comment:
http://www.hydrol-earth-syst-sci-discuss.net/hess-2016-272/hess-2016-272-AC3-supplement.pdf